# OpenReview forum: "Query-Dependent Prompt Evaluation and Optimization with Offline Inverse RL"
_ICLR.cc/2024/Conference — ICLR 2024 poster_

### Official Review · Reviewer_xV9W · 2023-10-30

**Soundness:** 3 good
**Presentation:** 2 fair
**Contribution:** 3 good
**Rating:** 6
**Confidence:** 4

**Summary:**

This paper proposes an “offline” (not using LLM as evaluators or prompt searchers) method to search effective “query-level” prompts. This method can potentially reduce the cost of prompt searching for enhancing the arithmetic reasoning ability of LLMs.

**Strengths:**

In the “offline” setting, this paper conducts relatively thorough experiments and ablations to find the optimal method.

**Weaknesses:**

However, my main concern is that it lacks a very important comparison with “on-line” methods. If the proposed method can achieve similar (or just a little bit lower) performance compared with “on-line” methods and it is significantly cheaper, it would be a good evidence that the proposed method can potentially be useful practically.

**Questions:**

See weakness.

---

> ### Author Response · Authors · 2023-11-15
> **Author Response to Reviewer xV9W**
>
> We deeply appreciate your time and effort devoted to reviewing our paper. And we are happy to have your acknowledgment of the significance and soundness of our work. Your insightful comments on the relationship between Prompt-OIRL and the online methods remind us to better present and contrast those approaches in our revision, and we have added a corresponding discussion section to our revised manuscript in Appendix D (New).**
>
> ---
>
>  In the following, we would start by noting that the online methods are the foundations for our query-dependent offline approach, and **additional prompts discovered by online algorithms can improve the performance of Prompt-OIRL.**
>
> 1. Additional prompt generated by the online method improves Prompt-OIRL’s performance upper bound.
>
> As the number of available prompts increases, the best-of-n performance among those prompts will be monotonically better. In addition to the evidence we have shown in the paper (Figure 6), the same conclusion can be drawn by comparing the performance upper bound of Prompt-OIRL on different tasks.
>
> In the table below, we use UB-Train-6 to denote the best performance (performance Upper-Bound) that can be achieved using the training prompts on the test queries, and use UB-Test-10 to denote the best performance that can be achieved using the 10 test prompts. Finally, we use UB-16 to denote the best performance that can be achieved using all those 16 prompts.
>
> | Dataset | LLM        | UB-Train-6 | UB-Test-10 | UB-16 |
> |---------|------------|--------------------------|-------------------------|--------------------------|
> | MAWPS   | GPT3.5-turbo | 0.942                | 0.941                  | 0.957                   |
> | SVAMP   | GPT3.5-turbo | 0.882                | 0.869                  | 0.903                   |
> | GSM8K   | GPT3.5-turbo | 0.872                 | 0.873                   | 0.914                    |
> | MAWPS   | LLaMA2-7B   | 0.891                 | 0.855                  | 0.926                   |
> | SVAMP   | LLaMA2-7B   | 0.876                 | 0.837                  | 0.907                   |
> | GSM8K   | LLaMA2-7B   | 0.582                  | 0.501                   | 0.648                    |
> | MAWPS   | TigerBot-13B | 0.929                | 0.642                  | 0.935                   |
> | SVAMP   | TigerBot-13B | 0.928                 | 0.913                 | 0.954                   |
> | GSM8K   | TigerBot-13B | 0.754                 | 0.7                     | 0.822                    |
>
> From the results, we can observe that the best performance can be achieved by the test prompts are usually worse than the best of the training prompts — this is not surprising given the fact that those training prompts are generated and optimized by experts or online algorithms. That said, we can also observe that selecting from the combination of all those 16 prompts can always lead to a significant improvement. Such an observation verifies the idea that adding additional prompts that are generated by the online prompts, either to the training set or to the test set, can be beneficial to Prompt-OIRL.
>
> 2. Monotonic Performance Improvement with the Number of Prompts.
>
> To add additional empirical evidence to our discussion, we can observe from Figure 6 in the main text that all methods, including all baseline methods and our proposed method, can have improved performance when more available prompts are available.
>
> The table below contextualizes the averaged performance improvement over different LLMs and tasks.
> We use “Improv. w/ +train” to highlight the improvement achieved using 6 training prompts as compared to only using 1 prompt. We use “Improv. w/ +test” to highlight the improvement achieved using additional test prompts as compared to only selecting from the training prompts (i.e., the improvement of Prompt-OIRL over BoTr Eqn(2)).
>
> | Method      | K = 1    | K = 2    | K = 3    | K = 4    | K = 5    | K = 6    | Improv. w/ +train |
> |-------------|----------|----------|----------|----------|----------|----------|--------------------|
> | LLM-Conf.   | 0.4751   | 0.5184   | 0.5427   | 0.5543   | 0.5594   | 0.5601   | + 17.89%           |
> | BoTr Eqn(1) | 0.4751   | 0.5379   | 0.5654   | 0.5800   | 0.5893   | 0.5944   | + 25.11%           |
> | BoTr Eqn(2) | 0.4751   | 0.5553   | 0.5910   | 0.6130   | 0.6295   | 0.6393   | + 34.56%           |
> | Prompt-OIRL | 0.5904   | 0.6095   | 0.6225   | 0.6331   | 0.6410   | 0.6459   | + 9.4%             |
> | Improv. w/ +test | +24.27% | +9.76% | +5.33% | +3.23% | +1.83% | +1.03% |          -          |
>
> **The monotonic improvement in all those comparisons demonstrates the benefit of integrating additional prompts, which can either be discovered through an existing online algorithm or crafted by human experts.**
>
> ---
>
> Once again, we thank the reviewer for their effort in improving our work. If there should be any remaining concerns or questions, we are keen to do our utmost to address them.

---

> ### Author Response · Authors · 2023-11-21
> **Further Discussions and Feedback Welcome**
>
> We deeply appreciate the insights you've shared during the review process. Following our revisions and previous responses, we are genuinely curious if we have adequately addressed the concerns you raised.
>
> We would appreciate it if you could kindly let us know if there were any further questions. In the extremely limited time remaining, we are still eager to do our utmost to address them!

---

> ### Author Response · Authors · 2023-11-22
> **Summarization of Feedback and Author Response**
>
> We extend our gratitude to the reviewer for their insightful feedback. As we approach the conclusion of the discussion period and have yet to receive further comments, we would like to summarize the initial feedback from the reviewer and our subsequent responses.
>
> ---
>
> In their initial review, the reviewer acknowledged the soundness of our contribution by conducting a thorough empirical study and detailed ablations in an offline setting. They recognized the effectiveness of our approach in utilizing query-level prompts to reduce costs and improve the arithmetic reasoning capabilities of LLMs. They raised a concern regarding the **connection between Prompt-OIRL and the online prompt-generation methods.**
>
> To address the reviewer’s concern about the link with the online methods,
>
> 1. Analytically,
> We emphasized that **Prompt-OIRL is not in direct competition with online methods but rather builds upon their generated knowledge**. This approach is akin to standing on the shoulders of giants; it improves upon the existing prompts by considering query dependency in test time, whether they are generated by online algorithms or domain experts. **This is evident in Figure 6, where an increase in the number of prompts correlates with consistent improvements in performance.** This demonstrates **Prompt-OIRL's capacity to draw valuable insights from online methods and exhibit monotonically better results.**
>
> 2. Synthetically, in our previous response, we posited that assuming an oracle reward model, the **incorporation of more prompts at test time consistently enhances the performance upper bound**. This holds true even if the additional prompts sourced from online methods are suboptimal. Thus, our study highlights the **importance of diverse prompting strategies**. This concept echoes our initial motivating example where different queries benefit from varying prompts, a finding supported in the concurrent literature [OPRO].
>
> 3. Empirically, we underpinned our reasoning with experimental evidence, demonstrating that:
>   - (1). Increasing the number of prompts sourced from online algorithms **during training consistently enhances the performance of Prompt-OIRL** and other baseline methods. This improvement **underscores the value of enriched demonstration data derived from online algorithms**.
>   - (2).  Increasing the number of prompts sourced from online algorithms **during testing leads to consistent improvements in query-dependent prompt optimization**. This is clearly observed when comparing the outcomes of Prompt-OIRL — which selects the most effective prompt from both training and testing sets using its learned reward model — against BoTr Eqn(2), which limits its selection to training prompts. The **performance gains observed in Prompt-OIRL affirm the benefits of integrating additional prompts from online sources.**
>
> ---
>
> We trust that the aforementioned clarifications sufficiently address the reviewer's concern. We are grateful for the opportunity to elaborate on these aspects and to underscore the importance of our findings in contributing to the field.
>
>
> **We would greatly appreciate any additional feedback from the reviewer on ways to further enhance the quality of our manuscript. Despite the extremely limited time remaining, we remain committed to doing our utmost to address any additional concerns or suggestions the reviewer may have.**
>
> ---
> **_Reference_**
>
> [OPRO]  Large Language Models as Optimizers (OPRO) Chengrun Yang, Xuezhi Wang, Yifeng Lu, Hanxiao Liu, Quoc V. Le, Denny Zhou, Xinyun Chen arXiv 2023.

---

### Official Review · Reviewer_QVHX · 2023-11-03

**Soundness:** 3 good
**Presentation:** 3 good
**Contribution:** 3 good
**Rating:** 6
**Confidence:** 4

**Summary:**

The authors propose a new method called Prompt-OIRL to improve the arithmetic reasoning abilities of large language models (LLMs) through zero-shot prompt optimization. They identify query dependency as an important objective in prompt optimization, but current techniques face two key challenges: 1) the lack of effective methods to evaluate prompts without access to the golden answers, and 2) the high computational cost of exploring the vast prompt space through interactions with the LLM. To address these issues, Prompt-OIRL utilizes offline inverse reinforcement learning on existing prompting benchmark data to learn a prompt evaluation model without needing the LLM. This model can then efficiently recommend optimal prompts in a best-of-N fashion for new queries. Experiments across various LLM scales and math datasets demonstrate Prompt-OIRL's efficacy and cost-effectiveness for zero-shot prompt optimization with query dependency.

**Strengths:**

1. The paper is well-structured and easy to follow
2. The idea is very interesting and the topic is important for automatic prompt engineering
3. Query-dependent evaluation is an essential challenge that is typically ignored. The authors identify this unique issue and solve it by proposing an RL framework, which looks very promising.
4. The experiments are strong enough to support the claims by comparing them with multiple SOTA baselines.

**Weaknesses:**

1.  While the method looks promising, I still expect to see potential discussion about the limitations, e.g., stability of inverse RL？
2. The scope is limited by arithmetic reasoning but the title seems a more generic framework that can be used to solve more broader tasks across different NLP tasks.
3. What's the current bottleneck if the proposed framework is applied to other instruction prompt optimization tasks, listed in ORPO[1], APE[2], and APO[3] baseline methods?
4. Without query dependence, what's the performance drop? Can you prove the necessity of that？
5. What's the current computational cost of the proposed framework？
6. Is that possible to compare with the GPT-4 or PaLM 2 model as well？


[1] Large Language Models as Optimizers (OPRO)
Chengrun Yang, Xuezhi Wang, Yifeng Lu, Hanxiao Liu, Quoc V. Le, Denny Zhou, Xinyun Chen
arXiv 2023.

[2] Automatic Prompt Optimization with "Gradient Descent" and Beam Search (APO)
Reid Pryzant, Dan Iter, Jerry Li, Yin Tat Lee, Chenguang Zhu, Michael Zeng
EMNLP 2023.

[3] Large Language Models Are Human-Level Prompt Engineers (APE)
Yongchao Zhou, Andrei Ioan Muresanu, Ziwen Han, Keiran Paster, Silviu Pitis, Harris Chan, Jimmy Ba
ICLR 2023.

**Questions:**

see weakness section

---

> ### Author Response · Authors · 2023-11-15
> **Author Response to Reviewer QVHX (1/3)**
>
> **We thank the reviewer for their insightful comments, and their acknowledgement of the soundness, presentation, and contribution of our paper. We aim to address all the individual points in your review here, but please also see the revised manuscript for changes (highlighted in red).**
>
> ---
>
> ### Q1. While the method looks promising, I still expect to see potential discussion about the limitations, e.g., stability of inverse RL？
>
> ### A1.
> Thank you for raising the concern about stability and robustness. In general, our offline-IRL approach enjoys high stability since the reward modeling step is based on gradient boosting methods and optimized through supervised learning. Similar to other offline-RL literature, learning with a fixed dataset also avoids the dilemma of exploration-exploitation trade-off which is usually challenging in the online approach. (e.g., in OPRO, APE, and APO [1-3]). In our work, we applied xgboost with a universal hyper-parameter setting to demonstrate the robustness of our approach.
>
> A temporary challenge for Prompt-OIRL’s wide application lies in the availability of open-accessible data. We will elaborate on this point with more details in A3 below.
>
> ---
>
> ### Q2. The scope is limited by arithmetic reasoning but the title seems a more generic framework that can be used to solve more broader tasks across different NLP tasks.
>
> ### A2.
> Thank you for the suggestion. To avoid potential misunderstanding, we have updated our title with a subtitle in our revision
> to better highlight the use case studied in our work. The updated title is
> > *Query-Dependent Prompt Evaluation and Optimization with Offline Inverse RL
> — A Case Study on Arithmetic Reasoning*

---

> > ### Author Response · Authors · 2023-11-15
> > **Author Response to Reviewer QVHX (2/3)**
> >
> > ### Q3. What's the current bottleneck if the proposed framework is applied to other instruction prompt optimization tasks, listed in ORPO[1], APE[2], and APO[3] baseline methods?
> >
> > ### A3.
> > Thank you for this question. We would answer this question by first providing our rationale in focusing on the arithmetic reasoning tasks and then highlighting a challenge and opportunity for the community — inspired by the success of Prompt-OIRL.
> >
> > 1. Motivation of centering on the arithmetic reasoning tasks
> >
> > The current scope of our work is centered on arithmetic reasoning tasks as the arithmetic reasoning ability is considered to be one of the emergent abilities of LLMs [4] that has “been essential yet challenging for language models and NLP models broadly” [5,6]. On the other hand, the rich demonstration data and expert knowledge in zero-shot prompting developed for arithmetic reasoning tasks enable accurate and rigorous study of the potential and effectiveness of the proposed method.
> >
> > 2. Challenge and opportunity: Accessibility of Offline Data - we would call on the community’s attention by demonstrating the value of those assets through the success of Prompt-OIRL
> >
> > To apply the key insight of Prompt-OIRL to more general natural language generation tasks, the challenge is on the **open-accessibility of the offline datasets.**
> >
> > Intuitively, one may think the challenge lies in the scarcity of prompting knowledge (i.e., the number of expert-crafted prompts). Yet as a matter of fact, an important observation we have made in our work is that in arithmetic reasoning datasets is that learning a reward model for query-dependent prompt evaluation and optimization **is helpful even in the scarce-expert knowledge setting**. Therefore, the scarcity of expert-crafted prompts is not a vital challenge.
> >
> > On the other hand, the accessibility of open-sourced offline datasets — under the form of (query, prompt, prompted response) — is more important. In the era of LLMs, only a few researchers release their prompted responses when releasing their discovered prompting strategies. Even having access to their code, to fully reproduce their results still require intensive interactions with the LLMs, leading to a huge amount of cost.
> >
> > Motivated by the discovery made by Prompt-OIRL, those offline datasets — not only those interactive communication logs with LLMs for the discovered well-performing prompts but also those logs for the failed prompting attempts — can be useful for learning a better reward model. Therefore, with the success of Prompt-OIRL, we would like to call on the community to release not only code but also the interaction logs (i.e., offline demonstration dataset) as an asset. Learning with those demonstrations saves energy and computation by avoiding repetition, and enhances the development of prompt engineering.
> >
> > **Actions taken** We have updated our manuscript and added this discussion to the appendix.
> >
> > ---
> >
> > **_References_**
> >
> > [4] Wei, Jason, et al. "Emergent abilities of large language models." arXiv preprint arXiv:2206.07682 (2022).
> >
> > [5] Blodgett, Su Lin, and Michael Madaio. "Risks of AI foundation models in education." arXiv preprint arXiv:2110.10024 (2021).
> >
> > [6] Rae, Jack W., et al. "Scaling language models: Methods, analysis & insights from training gopher." arXiv preprint arXiv:2112.11446 (2021).

---

> > > ### Author Response · Authors · 2023-11-15
> > > **Author Response to Reviewer QVHX (3/3)**
> > >
> > > ### Q4. Without query dependence, what's the performance drop? Can you prove the necessity of that？
> > > ### A4.
> > > We thank the reviewer for raising the question.
> > >
> > > Yes, the query-dependent objective is necessary and can be proved through our ablation studies. In our work, the improvement of query dependency can be empirically verified by comparing BoTr using Eqn(1), which is query-independent, and Eqn(2), which is query-dependent. Taking query-dependency into consideration, the performance on arithmetic reasoning tasks can be improved up to 7\%. The improvement over different demonstration availability is presented in Figure 6.
> > >
> > > ---
> > >
> > > ### Q5. What's the current computational cost of the proposed framework？
> > > ### A5.
> > > Thank you for the question.
> > > **Prompt-OIRL is highly computationally efficient. This is a huge advantage of our proposed offline inverse-RL approach.**
> > > As Prompt-OIRL does not rely on calling LLMs during optimization, its computational cost is significantly cheaper than the online methods. And Prompt-OIRL can be run on a laptop without GPU acceleration.
> > > With our implementation, training a reward model on a MacBook Air with an M2 Chip takes less than 20 minutes. When GPU servers are available, the computation can be further accelerated. Using a server with 16-core AMD 3995WX CPUs, the reward modeling step takes only 5 mins for each experiment setting.
> > >
> > > ---
> > >
> > > ### Q6. Is that possible to compare with the GPT-4 or PaLM 2 model as well？
> > >
> > > ### A6.
> > >
> > > Thank you for the question.
> > >
> > > We additionally experiment with GPT-4 on the MAWPS dataset. With a stronger language model, the reward modeling step of Prompt-OIRL will be more challenging, this is because of the increased imbalance of the data used for training the reward model (i.e., most prompts lead to a correct answer). That said, under such a more challenging setting, the efficacy of Prompt-OIRL and the query-dependent objective based on the learned reward model have been demonstrated.
> > >
> > > | Method      | K = 1       | K = 2       | K = 3       | K = 4       | K = 5       | K = 6       |
> > > |-------------|-------------|-------------|-------------|-------------|-------------|-------------|
> > > | BoTr Eqn(1) | 0.8781      | 0.9068      | 0.9131      | 0.9190      | 0.9245      | 0.9300      |
> > > | BoTr Eqn(2) | 0.8781      | 0.9352      | 0.9513      | 0.9609      | 0.9685      | 0.9757      |
> > > | Prompt-OIRL | 0.8944      | 0.9401      | 0.9526      | 0.9607      | 0.9672      | 0.9745      |
> > >
> > > As we have discussed in A3, the challenge of applying Prompt-OIRL to different settings is mainly on their data availability, which can be generated as by-product when researchers evaluating different prompts. Given the computational efficiency advantage of Prompt-OIRL, deploying our method on new tasks/LLMs will be smooth as long as the offline data is available.
> > >
> > > **Actions taken** We have added the new results to our updated manuscript. (Appendix E.3)
> > >
> > > ---
> > >
> > > Once again, we thank the reviewer for their effort in improving our work. If there should be any remaining concerns or questions, we are keen to do our utmost to address them.

---

> ### Author Response · Authors · 2023-11-21
> **Further Discussions and Feedback Welcome**
>
> We deeply appreciate the insights you've shared during the review process. Following our revisions and previous responses, we are genuinely curious if we have adequately addressed the concerns you raised.
>
> We would appreciate it if you could kindly let us know if there are any further questions. In the extremely limited time remaining, we are still eager to do our utmost to address them!

---

> > ### Comment · Reviewer_QVHX · 2023-11-23
> > **Thanks for your detailed response**
> >
> > Dear authors,
> >
> > Thanks very much for your detailed responses. I think they have addressed my concern so I maintain my score to support the acceptance of this work.

---

### Official Review · Reviewer_mPPy · 2023-11-03

**Soundness:** 2 fair
**Presentation:** 2 fair
**Contribution:** 3 good
**Rating:** 6
**Confidence:** 3

**Summary:**

This work proposes to optimize/choose prompts on a per-query basis to improve LLM performance on arithmetic reasoning tasks. They first identify two challenges towards this objective: a) ground truth labels are missing at inference time, making prompt evaluation challenging and b) repeated LLM interactions are costly. To overcome both of these challenges they propose Prompt-OIRL, an offline inverse reinforcement learning approach to learn a reward function per model and dataset, predicting the success of a  prompt query pair. They demonstrate empirically that this approach outperforms a range of baselines and can improve LLM performance.

**Strengths:**

* The query-dependent prompt optimization setting is a novel and promising direction.
* Prompt-OIRL is both more accurate and precise at assessing prompt success than LMSC for held-out and seen prompts.
* The evaluation covers a range of 3 different tasks and models of different sizes.

**Weaknesses:**

* Dataset generation requires a large number of model interactions for every new task and model, as no cross-task or -model generalization has been demonstrated.
* Many experimental details remain unclear or are only discussed in the appendix. E.g. the modelling of the proxy reward is not discussed at all in the main text, with the appendix suggesting a combination of an LLM embedding of prompt and query followed by an XGBT.
* It is unclear which single prompt was used for the scarce demonstration setting in Figure 5. Crucially, there, all baseline methods were limited to this single prompt, while Prompt-OIRL could choose any of the 6 considered prompts. Similarly, a comparison to always choosing each of the considered prompts and always choosing the best one (using an oracle) is missing but would be crucial to assessing performance
* The failures of Prompt-OIRL (e.g. GSM8K with LLaMA or TigerBot (Figures 20 and 21)) where it performs worse amongst all considered methods by a large margin are only shown in the appendix. Ironically this section is titled "REWARD MODELING: IMPLEMENTATION MATTERS", suggesting it should be discussed in the main text.
* While the use of LLMs for embedding computation of prompts is unproblematic when choosing the best-of-n (with n=6) fixed prompts, it might be prohibitively expensive for different policy optimization approaches requiring a substantially larger number of prompts to be embedded. This should be highlighted more prominently and not presented as a major advantage of this method.

**Questions:**

### Questions
1) Can you describe in detail the modeling of the proxy reward and conduct an ablation over the approaches mentioned but rejected in Appendix C.2
2) Can you report the performance of the (on average) best (on the test set) prompt and what this prompt is for the different models and datasets? Can you similarly report the performance of always choosing the best prompt for every query (corresponding to the setting where the proxy reward models the success flawlessly)?
3) How does Prompt-OIRL differ from  BoTr Eqn.2 when trained on all considered prompts? Its description suggests that also selects the best of n prompts but it achieves worse performance in Figure 6.
4) How does performance depend on the training data-set size given that much smaller datasets with gold-standard annotations will be available for many real-world applications?

### Comments
* The paper could benefit from a careful copywriting pass that addresses typos and grammatical errors while homogenizing the writing style.
* Figure 7 would benefit from a relative scale, especially for the "Cost with Commercial APIs". It generally seems slightly misleading to report improvements when 100 Prompts are used here while the rest of the paper considers 6 prompts. Further, it remains unclear if or how the cost of embedding the prompts and evaluating the proxy reward is considered here.

### Conclusion
The paper presents query-dependent prompt optimization as an interesting and novel approach to improving LLM performance. While multiple models and benchmarks are considered in the empirical evaluation, details remain unclear and some baselines are missing, eroding the confidence in the presented improvements. Combined with the missing details on the exact modeling of the proxy reward as well as any ablation of this key component, I am leaning toward rejecting this paper.

### Post Rebuttal Update
I have raised my score in response to the detailed rebuttal addressing my questions satisfactorily. I believe the paper could further benefit from incorporating some of these results, presented during the rebuttal.

---

> ### Author Response · Authors · 2023-11-15
> **Author Response to Reviewer mPPy (1/6)**
>
> Thank you for the insightful comments and your acknowledgment of the novelty and contribution of our work.
>
> In our initial manuscript, we focused more on illustrating the key insight of leveraging offline IRL in prompt evaluation and optimization, and deferred the implementation details into the appendix due to the space limit of the main text. As pointed out by the reviewer, we agree expanding those details can further enhance the understanding of our work. As a consequence, we have updated those sections with expanded details. We have also updated the paragraph on training-test prompt generation in our updated version to enhance clarity.
>
> **In the following, we aim to address all the individual points in your review. Please also see the revised manuscript for changes (highlighted in red). As because of your insightful feedback, the clarity of our work has been significantly improved.**

---

> > ### Author Response · Authors · 2023-11-15
> > **Author Response to Reviewer mPPy (2/6)**
> >
> > ### Q1. Can you describe in detail the modeling of the proxy reward and conduct an ablation over the approaches mentioned but rejected in Appendix C.2
> >
> > ### A1.
> >
> > Thank you for raising your concern and sharing your interest in the details of the reward modeling. In our initial draft, we deferred the detailed information about the offline dataset and the discussion on the reward model to the appendix due to space constraints in the main text. We have moved the key contents of reward modeling implementation into the main text to make it further self-contained.
> >
> > Moreover, we agree with the reviewer that providing a more detailed empirical comparison and analysis will be invaluable not only for enhancing the understanding of our method but may also contribute to the general community in understanding the learning with embeddings.
> > We have extended our discussion on reward modeling (in Appendix C) to include the following empirical evidence we observed in experiments.
> >
> >
> > 1. MLPs easily converge to the trivial classifiers
> >
> > For the MLP model, we have tried different choices of its hyper-parameters, including unit numbers, layers, various drop-out choices, and dual-channel architecture with each channel process query and prompt embeddings individually. However, we find all of those choices tend to converge to the trivial solution that predicts either all 0 or all 1 for the binary classification task. Such a reward model does not actually have the prediction ability in inference time. In practice, with such a reward mode, we can do nothing better than select the best-performing prompt on the training dataset. Therefore, the performance of BoTr Eqn.(1) represents the best-achievable performance of using a dummy MLP reward model.
> >
> >
> > 2. XGBoost is robust on hyper-parameter for reward modeling
> >
> > For the xgboost method, we set a universal hyper-parameter for all tasks and LLMs. When deploying Prompt-OIRL in practice, a case-by-case engineering on the hyper-parameters on reward modeling should be able to further boost the performance of the algorithm, yet this is out of our research scope. In our paper, we experiment with a single universal hyper-parameter setting for all LLMs and tasks, demonstrating the robustness of the proposed method.
> >
> >
> > 3. Instance-wise prediction of outcome is better than pair-wise prediction (preference-based learning)
> >
> > Another alternative for reward modeling is based on preference-based learning, which is effective in some conventional inverse RL tasks [1. T-REX]. To be specific, for every training query $x^{(i)}$, there may exist multiple prompts (denoted as $p_+^{(i)}$) that lead to a correct answer, and some other prompts ($p_-^{(i)}$) that lead to a wrong answer.
> >
> >
> > We can also organize the offline prompt demonstration in the form of preference-based dataset $(x^{(i)}, p_-^{(i)}, p_+^{(i)} )$, and learn from those pair-wise preference data. In this pair-wise preference approach, the learned reward model takes both prompts and the query as input, and outputs the preferred prompt. Given a new query in the inference time, such a reward model can be applied to all K candidate prompts with K-1 comparisons to find the best prompt. (so it is more computationally expensive than the direct reward modeling method used in our work). We empirically studied whether such an approach can lead to better performance on the MAWPS dataset with the GPT-3.5-turbo model. The results are shown in the table below:
> >
> >
> > | # Training Prompts | 1          | 2          | 3          | 4          | 5          | 6          |
> > |--------------------|------------|------------|------------|------------|------------|------------|
> > | BoTr Eqn(1)        | 0.6598     | 0.6988     | 0.7118     | 0.7155     | 0.7173     | 0.7175     |
> > | BoTr Eqn(2)        | 0.6598     | 0.7742     | 0.8150     | 0.8350     | 0.8473     | 0.8546     |
> > | BoTr Eqn(2) Paired | 0.6598     | 0.7195     | 0.7203     | 0.7194     | 0.7236     | 0.7247     |
> > | Prompt-OIRL        | 0.7637     | 0.8032     | 0.8379     | 0.8750     | 0.8916     | 0.8944     |
> >
> >
> > It can be concluded that the pair-wise reward model can not achieve better performance than the direct reward modeling used in Prompt-OIRL.
> >
> >
> > All our offline datasets and code for processing those datasets, as long as the MLP implementation and pair-wise reward modeling implementation will be made publicly available and contributed as an asset for future research.
> >
> >
> >
> >
> > **Actions taken** We have updated our manuscript accordingly to include extended details for the reward model training. Revisions have been highlighted in red in our updated manuscript.
> >
> >
> >
> > ---
> >
> > **_Reference_**
> >
> > [1. T-REX] Brown, Daniel, et al. "Extrapolating beyond suboptimal demonstrations via inverse reinforcement learning from observations." International conference on machine learning. PMLR, 2019.

---

> > > ### Author Response · Authors · 2023-11-15
> > > **Author Response to Reviewer mPPy (3/6)**
> > >
> > > ### Q2. Can you report the performance of the (on average) best (on the test set) prompt and what this prompt is for the different models and datasets? Can you similarly report the performance of always choosing the best prompt for every query (corresponding to the setting where the proxy reward models the success flawlessly)?
> > >
> > > ### A2.
> > > We thank the reviewer for pointing out those valuable performance references: those results can have reference value in understanding the improvement and future potential of Prompt-OIRL. We would also note that in practice, we will not be able to identify those best prompts according to their performance as the test label is not accessible.
> > >
> > > 1. Best test-set prompting strategy on different datasets:
> > > We use Best of Train to denote the best performance of training prompts (i.e., out of the 6 training prompts listed in Table 2 of our paper) on the test queries, and use Best of Test to denote the best performance of the first 10 held-out test prompts. Details of those test prompts are provided in Appendix C.3.
> > >
> > > | Dataset | LLM        | Best of Train | Train Prompt Index | Best of Test | Test Prompt Index |
> > > |---------|------------|---------------|--------------------|--------------|-------------------|
> > > | MAWPS   | GPT3.5-turbo | 0.855       | 0                  | 0.841       | 5                 |
> > > | SVAMP   | GPT3.5-turbo | 0.718       | 2                  | 0.736       | 2                 |
> > > | GSM8K   | GPT3.5-turbo | 0.672       | 1                  | 0.670       | 2                 |
> > > | MAWPS   | LLaMA2-7B   | 0.646        | 0                  | 0.587       | 8                 |
> > > | SVAMP   | LLaMA2-7B   | 0.637        | 0                  | 0.588       | 8                 |
> > > | GSM8K   | LLaMA2-7B   | 0.248        | 0                  | 0.241       | 5                 |
> > > | MAWPS   | TigerBot-13B | 0.536       | 0                  | 0.669       | 0                 |
> > > | SVAMP   | TigerBot-13B | 0.645       | 1                  | 0.673       | 8                 |
> > > | GSM8K   | TigerBot-13B | 0.3940       | 3                  | 0.393       | 7                 |
> > >
> > > 2. Assuming we have an oracle reward model (i.e., on test time, we assume we can always pick the prompt that achieves a correct answer), this should correspond to an empirical performance upper bound of Prompt-OIRL
> > >
> > > | Dataset | LLM        | Train Oracle | Test Oracle | Train + Test Oracle |
> > > |---------|------------|--------------------------|-------------------------|--------------------------|
> > > | MAWPS   | GPT3.5-turbo | 0.942                | 0.941                  | 0.957                   |
> > > | SVAMP   | GPT3.5-turbo | 0.882                | 0.869                  | 0.903                   |
> > > | GSM8K   | GPT3.5-turbo | 0.872                 | 0.873                   | 0.914                    |
> > > | MAWPS   | LLaMA2-7B   | 0.891                 | 0.855                  | 0.926                   |
> > > | SVAMP   | LLaMA2-7B   | 0.876                 | 0.837                  | 0.907                   |
> > > | GSM8K   | LLaMA2-7B   | 0.582                  | 0.501                   | 0.648                    |
> > > | MAWPS   | TigerBot-13B | 0.929                | 0.642                  | 0.935                   |
> > > | SVAMP   | TigerBot-13B | 0.928                 | 0.913                 | 0.954                   |
> > > | GSM8K   | TigerBot-13B | 0.754                 | 0.7                     | 0.822                    |
> > >
> > >
> > > Take-Away Messages:
> > >
> > > 1. The optimal prompt is highly relevant to the task and LLM used. In 3 out of the 9 settings, the optimal test-time prompting strategy exists in the test prompts, rather than the training prompts generated by previous research.
> > > 2. The oracle performance can be limited by the ability of LLMs.
> > > 3. Increasing the diversity of prompts improves the oracle performance.
> > >
> > > **Actions taken** We have updated our manuscript and added the results with the oracle reward model in Table 6 and Table 7.

---

> > > > ### Author Response · Authors · 2023-11-15
> > > > **Author Response to Reviewer mPPy (4/6)**
> > > >
> > > > ### Q3. How does Prompt-OIRL differ from BoTr Eqn.2 when trained on all considered prompts? Its description suggests that also selects the best of n prompts but it achieves worse performance in Figure 6.
> > > >
> > > > ### A3.
> > > > As the reward model of Prompt-OIRL can evaluate any given prompt, in test time, it can choose a prompt not only from the training set (i.e., out of the 6 training prompts), but also from the test set (in our experiments, we use 110 held-out test prompts, as detailed in Appendix C.3). The performance gain of Prompt-OIRL over BoTr Eqn.2 is based on the generalization ability of Prompt-OIRL on test prompts.
> > > >
> > > > The table below contextualizes the performance improvement averaged over different LLMs and tasks.  We use “OIRL/Eqn(2)” to highlight the improvement achieved using additional test prompts as compared to only selecting from the training prompts (i.e., the improvement of Prompt-OIRL over BoTr Eqn(2)).
> > > >
> > > >
> > > > | Method      | K = 1    | K = 2    | K = 3    | K = 4    | K = 5    | K = 6    |
> > > > |-------------|----------|----------|----------|----------|----------|----------|
> > > > | LLM-Conf.   | 0.4751   | 0.5184   | 0.5427   | 0.5543   | 0.5594   | 0.5601   |
> > > > | BoTr Eqn(1) | 0.4751   | 0.5379   | 0.5654   | 0.5800   | 0.5893   | 0.5944   |
> > > > | BoTr Eqn(2) | 0.4751   | 0.5553   | 0.5910   | 0.6130   | 0.6295   | 0.6393   |
> > > > | Prompt-OIRL | 0.5904   | 0.6095   | 0.6225   | 0.6331   | 0.6410   | 0.6459   |
> > > > | OIRL/Eqn(2) | +24.27% | +9.76% | +5.33% | +3.23% | +1.83% | +1.03% |
> > > >
> > > > The monotonic improvement demonstrate the benefit of integrating additional prompts in the test set.
> > > >
> > > > **Actions taken** We have updated our presentation to further clarify the usage of the test prompts, and further added additional discussion and Table 8 in our appendix to highlight this point.
> > > >
> > > > ---
> > > >
> > > > ### Q4. How does performance depend on the training data-set size given that much smaller datasets with gold-standard annotations will be available for many real-world applications?
> > > >
> > > > ### A4.
> > > >
> > > > Thank you for this suggestion on stress-testing Prompt-OIRL. We agree that benchmarking the performance of Prompt-OIRL under a limited number of gold-standard annotations can help understand its capability. To stress-test Prompt-OIRL under such data-scarcity settings, we experiment with the SVAMP dataset using a demonstration dataset from the GPT3.5-turbo and remove different proportions of training data. We compare results by removing [0.0, 0.3, 0.5, 0.7, 0.8, 0.9] of the training dataset, which has 15000 samples. The following table presents the results when using the 6 prompts. We compare BoTr Eqn.(1) and LLM confidence as the data-size-agnostic baselines, and BoTr Eq.(2) as a data-size-dependent baseline.
> > > >
> > > > | Dataset Size | 15000 | 10500 | 7500  | 4500  | 3000  | 1500  | BoTr Eqn.(1) | LLM Conf. |
> > > > |--------------|-------|-------|-------|-------|-------|-------|--------------|-----------|
> > > > | Prompt-OIRL  | 0.894 | 0.887 | 0.866 | 0.852 | 0.840 | 0.827 | 0.717        | 0.686     |
> > > > | BoTr Eqn.(2) | 0.847 | 0.831 | 0.814 | 0.803 | 0.794 | 0.782 | 0.717        | 0.686     |
> > > >
> > > >
> > > > From those results, it can be observed that Prompt-OIRL consistently achieves better performance than using the query-agnostic prompting strategies, using LLM confidence as baselines, or using Equation (2) and the reward model to choose from the training prompt.
> > > >
> > > > **Actions taken** We have conducted an additional empirical study on changing the dataset size in Prompt-OIRL. Results are added to our revised manuscript. (New: Appendix E.1)

---

> > > > > ### Author Response · Authors · 2023-11-15
> > > > > **Author Response to Reviewer mPPy (5/6)**
> > > > >
> > > > > ### Q5. While the use of LLMs for embedding computation of prompts is unproblematic when choosing the best-of-n (with n=6) fixed prompts, it might be prohibitively expensive for different policy optimization approaches requiring a substantially larger number of prompts to be embedded. This should be highlighted more prominently and not presented as a major advantage of this method.
> > > > >
> > > > > ### A5.
> > > > >
> > > > > Using the embedding models to get prompt or query embeddings is in general much cheaper than calling the LLMs in getting responses.
> > > > > For instance, the gpt-3.5-turbo API charges 0.001 USD per 1k tokens for input, and 0.002 USD for output; as a comparison, the embedding model of ada v2 charges only 0.0001 USD per 1k token.
> > > > >
> > > > > Moreover, we would like to note those prompt embeddings are re-usable. For instance, getting the embedding for 100 prompts like “let’s try to solve it step by step, and make sure to get the correct answer” will cost about 100 prompts *  20 tokens/per prompt * 0.0001 USD / 1000 tokens = 0.0002 USD.
> > > > >
> > > > > In comparison, **interaction with the LLMs to get the prompted response is much more expensive.** A single query-answer pair will lead to a cost of approximately 0.0006 USD. Evaluating the performance of a single prompting strategy by interacting with gpt-3.5-turbo on a medium-sized dataset containing 5000 samples will lead to a 3 USD expense.
> > > > >
> > > > > Therefore, the cost of Prompt-OIRL in generating embeddings ($\approx$0.0002 USD) is significantly cheaper than the cost of evaluating any prompt with LLMs ($\approx$3 USD). The gap in cost will be even more significant when multiple prompts need to be evaluated (i.e., evaluating K different prompts will cost $\approx$3K USD).
> > > > >
> > > > > **Actions taken** We have added an additional section of discussion in our manuscript to discuss the expenses of embedding. (New: Appendix C.7)

---

> > > > > > ### Author Response · Authors · 2023-11-15
> > > > > > **Author Response to Reviewer mPPy (6/6)**
> > > > > >
> > > > > > ### Q6. Generalization over tasks, and requirements on data.
> > > > > >
> > > > > > ### A6.
> > > > > > Through the empirical results in our main text and A4, we demonstrated that Prompt-OIRL improves over baselines in both limited training data size settings and scarce expert-crafted prompting strategy settings. In this section, we follow the reviewer’s suggestion to further investigate the generalization ability of Prompt-OIRL over different tasks.
> > > > > >
> > > > > >
> > > > > > 1. Prompt-OIRL achieves promising results in when generalized over tasks
> > > > > >
> > > > > > We agree with the reviewer that the generalization of Prompt-OIRL over tasks can have great value in practical deployments. To verify the generalization ability of the learned reward model, we experiment by using reward models that are trained on different tasks in Prompt-OIRL when evaluating and optimizing prompts for the MAWPS task with GPT3.5-turbo. The following table shows the results we get using the three different reward models trained with MAWPS, SVAMP, and GSM8K, individually.
> > > > > >
> > > > > > | Model  | Method      | K = 1  | K = 2  | K = 3  | K = 4  | K = 5  | K = 6  |
> > > > > > |--------|-------------|-------|-------|-------|-------|-------|-------|
> > > > > > | MAWPS  | BoTr Eqn(1) | 0.6598 | 0.6988 | 0.7118 | 0.7155 | 0.7173 | 0.7175 |
> > > > > > |        | BoTr Eqn(2) | 0.6598 | 0.7742 | 0.8150 | 0.8350 | 0.8473 | 0.8546 |
> > > > > > |        | Prompt-OIRL | 0.7637 | 0.8032 | 0.8379 | 0.8750 | 0.8916 | 0.8944 |
> > > > > > | SVAMP  | BoTr Eqn(1) | 0.6598 | 0.6959 | 0.7078 | 0.7101 | 0.7124 | 0.7163 |
> > > > > > |        | BoTr Eqn(2) | 0.6598 | 0.7367 | 0.7529 | 0.7587 | 0.7593 | 0.7626 |
> > > > > > |        | Prompt-OIRL | 0.7222 | 0.7497 | 0.7757 | 0.7804 | 0.7883 | 0.7903 |
> > > > > > | GSM8K  | BoTr Eqn(1) | 0.6598 | 0.6753 | 0.6922 | 0.6968 | 0.6967 | 0.6926 |
> > > > > > |        | BoTr Eqn(2) | 0.6598 | 0.6880 | 0.6982 | 0.7016 | 0.7007 | 0.7092 |
> > > > > > |        | Prompt-OIRL | 0.6617 | 0.6884 | 0.6985 | 0.7015 | 0.7012 | 0.7145 |
> > > > > >
> > > > > > It is worth noting that, under such a cross-task setting, the BoTr Eqn(1) baseline selects the best prompt according to the training set performance on the training dataset of the reward model. For instance, when generalizing the reward model learned from the GSM8K prompting demonstration dataset to the MAWPS task, its Best-of-Training prompt is chosen to be the best-performing prompt on the GSM8K dataset.
> > > > > >
> > > > > > According to the results, we find the reward model learned with the SVAMP dataset generalizes better to the MAWPS task than the reward model learned with the GSM8K dataset, this is not surprising as those two tasks are more similar as compared to the GSM8K dataset which always requires multiple step reasoning. Importantly, the performance of Prompt-OIRL consistently surpasses using the best-of-training baselines, demonstrating its potential in practical deployments where generalization over tasks is considered to be an essential advantage.
> > > > > >
> > > > > >
> > > > > >
> > > > > > 2. Generalization over LLM requires a strong assumption
> > > > > >
> > > > > > Following the common practice in RLHF, where reward models are also used, Prompt-OIRL uses LLM-specific reward models in prompt evaluation. This is due to the fact that different language models are often trained with different model architectures, training settings, and data. On the other hand, generalizing the learned reward model across language models assumes the prompting mechanism is agnostic to all of those differences, which is unlikely to hold. Empirically, the recent paper of OPRO [2] has shown evidence that the optimal prompting strategies for different language models are different.
> > > > > >
> > > > > > We thank the reviewer again for their insightful comments, as because of the promising results in cross-task generalization, the potential impact and applicability of our proposed method is greatly improved.
> > > > > >
> > > > > > **Actions taken** We have added additional experiments studying the generalization of Prompt-OIRL over tasks (Appendix E.2). The promising results demonstrate the potential of Prompt-OIRL in real-world deployments.
> > > > > >
> > > > > >
> > > > > >
> > > > > > **_Reference_**
> > > > > >
> > > > > > [2. OPRO] Yang, Chengrun, et al. "Large language models as optimizers." arXiv preprint arXiv:2309.03409 (2023).
> > > > > >
> > > > > > ---
> > > > > >
> > > > > > Once again, we thank the reviewer for their effort in improving our work. If there should be any remaining concerns or questions, we are keen to do our utmost to address them.

---

> ### Comment · Reviewer_mPPy · 2023-11-21
> **Thank you for the Detailed Rebuttal**
>
> I want to thank the authors for going to great lengths to provide a detailed rebuttal, addressing all my questions and concerns. I believe especially the results on generalization across datasets and strong performance even on small datasets to be promising signs for the practical applicability of this approach.
>
> I have thus raised my score.
>
> I would still encourage the authors to address the following points:
> * Highlight the fact that they use an LLM generated embedding as input to their XGBT in Section 3 Step 2.
> * Include a direct comparison of oracle performance with their approach and BoTr Eqn. 1 to better contextualize the performance improvement realized by their method.
> * Update Figure 7 with relative scales.

---

> > ### Author Response · Authors · 2023-11-21
> > **Author Response**
> >
> > Thank you for your follow-up feedback.
> >
> > We greatly appreciate your acknowledgment of our efforts to address the previous concerns and are encouraged by your subsequent re-evaluation of our work. We agree that the implementation of the recommended experiments, specifically on cross-task generalization and stress tests for small dataset sizes, has substantially validated the practical applicability of Prompt-OIRL.
> >
> > We have made additional revisions to address the three mentioned points:
> > 1. We highlighted the usage of embeddings explicitly in Section.3, Step 2.
> > 2. We include the oracle score as the normalizer to better contextualize the performance gain and future potential of Prompt-OIRL.
> > 3. We have updated Figure 7 and used an updated relative scale for cost on the y-axis.
> >
> > Once again, we deeply appreciate your efforts and insightful suggestions in improving our paper, as because of it, the significance,  contribution, and impact of our work are more effectively highlighted. Please let us know if there are any leftover concerns, we are keen to further address them.

---

> > > ### Comment · Reviewer_mPPy · 2023-11-21
> > > **Wrong PDF version?**
> > >
> > > I want to thank the authors for being so responsive and actively engaging in the discussion
> > >
> > > It seems like the changes to Section 3 Step 2 present in the previous revision got lost in the current version. Similarly, I did not spot the additional information on using embeddings. It also seems that Figure 7 is unchanged.
> > >
> > > I am thus wondering if they perhaps uploaded an incorrect pdf.

---

> > > > ### Author Response · Authors · 2023-11-21
> > > > **Author Response**
> > > >
> > > > Thanks for your prompt feedback on the version issue. We have uploaded our latest manuscript.
> > > >
> > > > In the latest version, we have replaced the previous Figure 7 with a table to better contrast the cost-efficiency gap between Prompt-OIRL and the LLM-based method. We take the cost of embedding into consideration in this updated table and explicitly mention the embedding in the updated caption.
> > > >
> > > > Below please find a preview of the updated table:
> > > >
> > > > | Model         | Cost         | 1 Prompt | 6 Prompts | 110 Prompts |
> > > > |---------------|----------------|----------|-----------|-------------|
> > > > | **LLaMA2-7B** | LLM-Based      | 0.00325  | 0.01949   | 0.35723     |
> > > > |               | Prompt-OIRL     | 0.00134  | 0.00134   | 0.00134     |
> > > > | **TigerBot-13B**| LLM-Based    | 0.00082  | 0.00492   | 0.09029     |
> > > > |               | Prompt-OIRL     | 0.00034  | 0.00035   | 0.00056     |
> > > > | **GPT-3.5-turbo**| LLM-Based   | 0.00093  | 0.00558   | 0.10237     |
> > > > |               | Prompt-OIRL     | 0.00040  | 0.00041   | 0.00062     |
> > > >
> > > > ---
> > > >
> > > > Once again, we deeply appreciate the insights you've shared during the review process. Please let us know if you have any remaining questions or comments, we will do our utmost to address them!

---

### Official Review · Reviewer_wFdE · 2023-11-04

**Soundness:** 4 excellent
**Presentation:** 3 good
**Contribution:** 3 good
**Rating:** 6
**Confidence:** 3

**Summary:**

This paper aims to enhance the arithmetic reasoning ability of LLM via prompt optimization. Inspired by the fact that no prompt is perfect for all queries and existing online evaluations of different prompt choices are expensive, this work proposes an offline RL based prompt optimization solution. Given the existing human crafted prompts for different arithmetic reasoning datasets, a reward model without depending on LLM is trained to approximate the prediction by using LLM. Experimental evaluation on multiple arithmetic reasoning dataset with 3 different LLMs shows a strong performance.

**Strengths:**

This manuscript addresses a compelling issue, namely the optimization of query-dependent prompts, which is becoming increasingly relevant as Large Language Models (LLMs) see wider application across various contexts. Developing an efficient prompt strategy tailored to individual queries, without relying on costly LLM inference, is a pertinent and significant challenge.

The method put forward is both technically robust and effectively articulated. The authors have offered a comprehensive account of their approach, with a few exceptions (see below) The empirical outcomes presented are robust and offer a positive indication of the method's potential.

Moreover, the analysis and ensuing comparison with pertinent literature succinctly underscore the advantages and novel contributions of this research.

**Weaknesses:**

How does the proposed method perform when the offline prompt-alignment dataset is small? It is encouraging to observe that the method shows promise with the 10 held-out prompts and an expanded set of 100 prompts. Nevertheless, in real-world scenarios, we may encounter new tasks with a limited number of available prompts for offline data. I am curious about the method's performance across various quantities of training prompts.



Several critical technical details are absent from the main text. For example, there is little to no information about the curated offline dataset or the design principles behind the parameterized proxy reward model, among others.



Following these points, it's also vital to explore and articulate the different design choices for reward models. It is mentioned that an MLP model is less effective, yet a detailed analysis would be invaluable, assisting the reader to understand and tailor the method to their specific use cases.



Regarding the proxy reward model, a simpler, more straightforward query-dependent model, such as a nearest-neighbor based solution, could be considered. This would involve, for each query, locating the closest match whose prompt yields a correct answer and utilizing that prompt for the new query. Please consider incorporating this simpler solution as a baseline for comparison.

**Questions:**

What are the values of K, M and P in appendix section C.1?

---

> ### Author Response · Authors · 2023-11-15
> **Author Response to Reviewer wFdE (1/3)**
>
> **We thank the reviewer for the insightful comments and for their acknowledgment of the contribution, novelty, and significance of our work. We aim to address all the individual points in your review here, but please also see the revised manuscript for changes (highlighted in red).**
>
> ---
>
> ### Q.1 How does the proposed method perform when the offline prompt-alignment dataset is small?
> ### A.1
> Thank you for highlighting this concern. We agree that the applicability of our idea in settings when demonstration data is scarce (i.e., when the knowledge of effective prompting strategy is limited) is important.
>
>
> In our experiments, we consider the empirical setting when having access to a different number of expert prompting demonstrations. Specifically, the **scarce demonstration setting** presented in Figure 5 (left panel) experiments with only 1 (out of the 6) training prompt.
>
>
> In such a setting, as there is only a single expert prompt available during training time, the best-of-training time prompt (BoTr) strategy must choose to use this training prompt, and so does the LLM-Confidence. Differently, with a reward model training using the demonstration data collected from a single expert-crafted prompt, Prompt-OIRL can generalize its evaluation ability on held-out prompts and achieve significantly improved performance.
>
>
> In Figure 6, we change the availability of training prompts by the x-axis. We can observe that the performance improvement over baseline methods is most significant when only a single training prompt is used. Not surprisingly, when the number of human-crafted prompts increases, the best-of-training performance is greatly improved.
>
>
> Therefore, in practice, Prompt-OIRL shines even with a limited number of demonstration prompts, proving its great potential to be applied to settings when expert prompting knowledge is relatively scarce.
>
>
> > **Actions taken: We’ve updated our manuscript and improved the clarity in conveying such a point. We appreciate the reviewer for commenting on this point, as because of it, the clarity of the significance and general applicability of our work has been enhanced. Revisions have been highlighted in red in our updated manuscript.**

---

> ### Author Response · Authors · 2023-11-15
> **Author Response to Reviewer wFdE (2/3)**
>
> ### Q.2 Details of the offline dataset and the reward model.
> ### A.2
> We thank the reviewer for pointing out the importance of reward modeling details. Due to space constraints in the main text, we deferred the detailed information about the offline dataset and the discussion on the reward model to the appendix. To make our main text better self-contained, we have revised our manuscript and provided the essential empirical implementation of the reward model in the main text.
>
>
> In addition, we agree with the reviewer that providing a more detailed empirical comparison and analysis will be invaluable not only for enhancing the understanding of our method but may also contribute to the general community in understanding the learning with embeddings. Therefore, we have also updated our appendix and provided more details on the reward modeling to including the following empirical evidence:
>
>
> 1. MLPs easily converge to the trivial classifiers
>
> For the MLP model, we have tried different choices of its hyper-parameters, including unit numbers, layers, various drop-out choices, and dual-channel architecture with each channel process query and prompt embeddings individually. However, we find all of those choices tend to converge to the trivial solution that predicts either all 0 or all 1 for the binary classification task. Such a reward model does not have the prediction ability in inference time. In practice, with such a reward mode, we can do nothing better than select the best-performing prompt on the training dataset. Therefore, the performance of BoTr Eqn.(1) represents the best-achievable performance of using a dummy MLP reward model.
>
>
> 2. XGBoost is robust on hyper-parameter for reward modeling
>
> For the xgboost method, we set a universal hyper-parameter for all tasks and LLMs. When deploying Prompt-OIRL in practice, a case-by-case engineering on the hyper-parameters on reward modeling should be able to further boost the performance of the algorithm, yet this is out of our research scope. In our paper, we experiment with a single universal hyper-parameter setting for all LLMs and tasks, demonstrating the robustness of the proposed method.
>
>
> 3. Instance-wise prediction of outcome is better than pair-wise prediction (preference-based learning)
>
> Another alternative for reward modeling is based on preference-based learning, which is effective in some conventional inverse RL tasks [1. T-REX]. To be specific, for every training query $x^{(i)}$, there may exist multiple prompts (denoted as $p_+^{(i)}$) that lead to a correct answer, and some other prompts ($p_-^{(i)}$) that lead to a wrong answer.
>
>
> We can also organize the offline prompt demonstration in the form of preference-based dataset $ (x^{(i)} $ , $p_-^{(i)}$, $p_+^{(i)} )$, and learn from those pair-wise preference data. In this pair-wise preference approach, the learned reward model takes both prompts and the query as input, and outputs the preferred prompt. Given a new query in the inference time, such a reward model can be applied to all K candidate prompts with K-1 comparisons to find the best prompt. (so it is more computationally expensive than the direct reward modeling method used in our work). We empirically studied whether such an approach can lead to better performance on the MAWPS dataset with the GPT-3.5-turbo model. The results are shown in the table below:
>
>
> | # Training Prompts | 1          | 2          | 3          | 4          | 5          | 6          |
> |--------------------|------------|------------|------------|------------|------------|------------|
> | BoTr Eqn(1)        | 0.6598     | 0.6988     | 0.7118     | 0.7155     | 0.7173     | 0.7175     |
> | BoTr Eqn(2)        | 0.6598     | 0.7742     | 0.8150     | 0.8350     | 0.8473     | 0.8546     |
> | BoTr Eqn(2) Paired | 0.6598     | 0.7195     | 0.7203     | 0.7194     | 0.7236     | 0.7247     |
> | Prompt-OIRL        | 0.7637     | 0.8032     | 0.8379     | 0.8750     | 0.8916     | 0.8944     |
>
>
> It can be concluded that the pair-wise reward model can not achieve better performance than the direct reward modeling used in Prompt-OIRL.
>
>
> All our offline datasets and code for processing those datasets, as long as the MLP implementation and pair-wise reward modeling implementation will be made publicly available and contributed as an asset for future research.
>
>
> > **Actions taken: We have updated our manuscript accordingly to include extended details for the reward model training. Revisions have been highlighted in red in our updated manuscript.**
>
> ---
>
> **_Reference_**
>
> [1. T-REX] Brown, Daniel, et al. "Extrapolating beyond suboptimal demonstrations via inverse reinforcement learning from observations." International conference on machine learning. PMLR, 2019.

---

> ### Author Response · Authors · 2023-11-15
> **Author Response to Reviewer wFdE (3/3)**
>
> ### Q.3 A Nearest-Neighbor Baseline
> ### A.3
> We thank the reviewer for pointing out the nearest-neighbor method as an additional practical baseline. We would begin with a discussion on the differences between such a potential baseline and Prompt-OIRL, followed by experiment results.
>
>
> 1. The Nearest-Neighbor method can not be generalized to new prompts. The idea of using a nearest-neighbor approach can be an alternative to using a parameterized model in selecting from the training prompts. However, it can not be generalized to unseen prompts due to a lack of support.
>
>
> 2. As a non-parametric method, the Nearest-Neighbor approach requires memorization of the training embeddings, which can be expensive when the number of demonstrations increases.
>
>
> Empirically, we implemented this idea with GPT-3.5-turbo on the MAWPS dataset. To be specific, for every test query, we look up the training queries’ embedding to find the closest neighbor of this test query’s embedding, and select from the training prompt(s) that has successfully prompted correct answers on such a training query.
>
>
> The results are shown in the table below:
>
>
> | # Training Prompts | 1          | 2          | 3          | 4          | 5          | 6          |
> |--------------------|------------|------------|------------|------------|------------|------------|
> | BoTr Eqn(1)        | 0.6598     | 0.6988     | 0.7118     | 0.7155     | 0.7173     | 0.7175     |
> | BoTr Eqn(2)        | 0.6598     | 0.7742     | 0.8150     | 0.8350     | 0.8473     | 0.8546     |
> | Nearest-Neighbor   | 0.6598     | 0.7856     | 0.8242     | 0.8339     | 0.8492     | 0.8530     |
> | Prompt-OIRL        | 0.7637     | 0.8032     | 0.8379     | 0.8750     | 0.8916     | 0.8944     |
>
>
> The nearest-neighbor approach, as a query-denpendent baseline, outperforms the BoTr Eqn(1) baseline, and achieves on-par performance as BoTr Eqn(2). However, it underperforms Prompt-OIRL as the generalization ability of the learned reward model in Prompt-OIRL further enhanced its prompting performance.
>
>
> > **Actions taken: We analyzed the potential and challenge of applying the nearest neighbor method as an alternative approach to parametric models used in Prompt-OIRL, and provided additional experiment results to verify the idea.**
>
> ---
>
> ### Q.4 What are the values of K, M and P in appendix section C.1?
> ### A.4
> In Appendix C.1, N is the number of samples (query-answer pairs) in the training dataset, M is the number of samples in the held-out test dataset, K is the number of training prompts, and P is the number of test prompts.
>
>
> For all datasets, we experiment with different choices of K = [1, 2, 3, 4, 5, 6]. (i.e., the x-axis of Figure 6: number of training prompts, ranging from 1 - 6). And use a total number of P=110 held-out prompts.
>
>
> For GSM8K, there are N=7473 samples used for training, and M=1319 samples for testing.
> For SVAMP, there are N=15000 samples used for training, and M=4690 samples for testing.
> For MAWPS, there are N=6000 samples used for training, and M=1685 samples for testing.
>
>
> > **Actions taken: We have updated our description in Appendix C.1 to enhance the clarity. Our revision is highlighted in red.**
>
>
> ---
> Once again, we thank the reviewer for their effort in improving our work. If there should be any remaining concerns or questions, we are keen to do our utmost to address them.

---

> > ### Comment · Reviewer_wFdE · 2023-11-22
> > **Thank you for the response.**
> >
> > Thank you for your thorough response. I will maintain my positive rating. Although the nearest neighbor method is simple and possesses inherent limitations, as discussed, it remains an essential baseline for analysis. I recommend that the authors incorporate it into the final version of the paper.

---

> > > ### Author Response · Authors · 2023-11-22
> > > **Thank you for your feedback**
> > >
> > > We sincerely thank the reviewer for the additional feedback and for the encouraging positive affirmation of our work.
> > >
> > >
> > > Aligned with your suggestion, we appreciate the importance of integrating the Nearest-Neighbor baseline to facilitate a more comprehensive comparison and analysis. Accordingly, we have updated our manuscript to include the discussion on the Nearest-Neighbor baseline. This can be found in Appendix C.2, under the section of reward modeling details, on page 23-24 and Table 6, of the latest manuscript.
> > >
> > >
> > > Once again, we extend our gratitude to the reviewer for their invaluable insights, which have substantially contributed to the enhancement of our manuscript's quality. We remain open to and appreciative of any further suggestions or comments. We would be grateful if the reviewer might consider reassessing their rating or confidence score, taking into account the recent enhancements made to our paper.

---

> ### Author Response · Authors · 2023-11-21
> **Further Discussions and Feedback Welcome**
>
> We deeply appreciate the insights you've shared during the review process. Following our revisions and previous responses, we are genuinely curious if we have adequately addressed the concerns you raised.
>
> We would appreciate it if you could kindly let us know if there are any further questions. In the extremely limited time remaining, we are still eager to do our utmost to address them!

---

### Author Response · Authors · 2023-11-15
**Summary of Author Response**

We extend our sincere gratitude to all reviewers for their insightful comments, valuable suggestions, time, and efforts in evaluating and improving our paper.

We are very happy that the following aspects of our work were acknowledged by our reviewers:
- **Significance of the Problem**: [wFdE” a pertinent and significant challenge”, mPPy ”setting is a novel and promising”, QVHX ”topic is important”]
- **Writing and Presentation**: [wFdE, QVHX “well-structured and easy to follow”]
- **Novelty and Contribution**: [wFdE ”technically robust and effectively articulated”, mPPy, QVHX “idea is very interesting”]
- **Empirical Evaluation**: [wFdE ”comprehensive”, QVHX “experiments are strong enough to support the claims”, xV9W “this paper conducts relatively thorough experiments and ablations”]
---

Thanks to the insightful comments we received from the reviewers, we are able to further improve the clarity of our manuscript and enhance its quality. To address the concerns raised by reviewers, we would respond to each of their questions respectively. Below, as a general response, we aim to outline the **key revisions and additional experimentation conducted by far:**

**Supplementary Experimental Evaluation**

1. Added a **new baseline using the nearest neighbor method** as suggested by Reviewer wFdE
2. Added a **new experiment studying the effect of changing dataset size** on Prompt-OIRL as suggested by Reviewer mPPy
3. Added a **new experiment studying the generalization ability of Prompt-OIRL over tasks** as suggested by Reviewer mPPy
4. Added a **new experiment with GPT4** as suggested by Reviewer QVHX

**Revised Manuscript for Clarity**

1. We’ve added a subtitle *--a case study on arithmetic reasoning*, to better anchor our demonstrated application.
2. We provided more detail on reward modeling, including
  - 2.1 An update in Section 3, to make our main text better self-contained.
  - 2.2 An extended explanation of notations in Appendix C.1, to enhance clarity.
  - 2.3 A discussion on three different choices in reward modeling in Appendix C.2, showing the setbacks in MLP and preference-based learning.
3. We clarified our presentation in Section 5. Including the usage of test prompts, and the illustration of results, to better highlight the strong empirical performance of Prompt-OIRL under both prompt-scarce setting and rich demonstration setting.
4. We extended our discussion on future work, potential challenges, and link to the online methods in Appendix D. We hope to use the success of Prompt-OIRL to call on the community to pay attention to the valuable asset of LLM interactive logs, and contribute by releasing their offline prompt demonstration.

**In our updated manuscripts, all those changes are highlighted in red.**

---

Once again, we deeply appreciate the insightful comments and suggestions from our reviewers. With those comments addressed and corresponding revisions made to the manuscript, we believe the quality of our manuscript is further enhanced.

We hope our clarification and additional empirical studies could address the concerns raised by reviewers. Should there be any leftover questions, please let us know and we will make every effort to address them during the subsequent discussion period.

---

> ### Author Response · Authors · 2023-11-23
> **Post-Rebuttal Updates**
>
> We are deeply appreciative of all the reviewers for their insightful feedback and constructive comments, which have been pivotal in enhancing our work. By diligently addressing the concerns raised by the reviewers, we are confident that the quality of our paper has been substantially improved.
>
>
> **We are particularly thankful for the additional feedback provided by our reviewers during the author-reviewer discussion period.**
> **We are encouraged to see all of our reviewers acknowledge our response in addressing their initial concerns and have reached a consensus on the acceptance of our work.**
>
> ---
>
> During the discussion period, some of our reviewers provided additional suggestions, which have been invaluable in further refining our manuscript. In addition to the revisions listed above, we have also **implemented the following updates**
>
> 1. We have elaborated on the usage of embeddings in Section 3 to enhance clarity and understanding, following the suggestion of reviewer mPPy. _(page 5)_
> 2. We have integrated Figure 6 with the oracle baseline, to more effectively illustrate the performance gains and the prospective impact of Prompt-OIRL, as suggested by reviewer mPPy. _(page 7)_
> 3. The original Figure 7 has been replaced with a table, offering a clearer representation of Prompt-OIRL's cost-efficiency across various settings, as per the suggestion of reviewer mPPy. _(page 9)_
> 4. We have incorporated a discussion on the Nearest-Neighbor approach as an additional alternative for reward modeling. This inclusion aims to ensure the completeness of our study and to facilitate ease in future analyses, as recommended by reviewer wFdE. _(page 23-24)_
> 5. Our discussion on the interplay between Prompt-OIRL and online approaches has been expanded in Appendix D, inspired by the insights from reviewer xV9W. _(page 29-30)_
>
> **All of those revisions are highlighted in blue in our latest manuscript.**
>
> ---
> **Finally, we would like to highlight the contributions of Prompt-OIRL.**
>
> Our work presents four key **technical contributions**:
>
> 1. Formally, we identify the overlooked query-dependent prompt optimization objective and its challenges, and introduce Offline Inverse Reinforcement Learning to integrate rich human expertise as a systematic approach.
> 2. Methodologically, we introduce Prompt-OIRL to first perform a query-dependent offline prompt evaluation with a learned reward model and then perform offline prompt optimization to improve prompting performance.
> 3. Practically, we highlight the existence of offline datasets generated --- as by-products when existing prompting strategies are benchmarked on open-access tasks --- can be directly adopted for Prompt-OIRL learning.
> 4. Empirically, we validate the *efficacy* and *efficiency* of Prompt-OIRL in offline prompt evaluation and optimization through experiments with 3 distinct LLMs, namely GPT-3.5-turbo, LLaMA-2-7B-Chat, and TigerBot-13B-Chat, across 3 arithmetic datasets: GSM8K, MAWPS, and SVAMP.
>
> Furthermore, our work also **contributes to the community by highlighting the (overlooked) importance of offline LLM-interaction datasets in the era of LLM research.**
> - The significant performance improvement of Prompt-OIRL over the baselines demonstrates the efficacy and efficiency of learning with those offline demonstration datasets. Therefore, **we advocate for the broader LLM research community not only to release their code but also their interactive logs with LLMs.** This practice will enhance reproducibility in research and foster further applications, as demonstrated in our work.
>
>
>
> ---
> In closing, we wish to express our deepest gratitude to the reviewers for their thorough and thoughtful engagement with our work.
> We remain committed to advancing the quality of our paper, and welcome any additional suggestions or feedback.
>
> **Thank you once again for your invaluable contributions to our work.**

---

### Author Response · Authors · 2023-11-20
**Further Discussions and Feedback Welcome**

Dear Reviewers,

We deeply appreciate the insights you've shared during the review process. Following our revisions and previous responses posted 5 days ago, we are genuinely curious if we have adequately addressed the concerns you raised.


Should there be any leftover questions, concerns, or areas you feel need more clarification, please do not hesitate to let us know. We greatly respect your insights and stand ready to make any additional refinements based on your feedback.


Regards,

Submission 993 Authors

---

### Meta-Review · Area_Chair_U61X · 2023-12-05

**Metareview:**

The paper introduces an approach to address the query-dependent prompt optimization problem for enhancing arithmetic reasoning ability in LLMs. The Prompt-OIRL method, employing offline inverse reinforcement learning, demonstrates promising results, outperforming various baselines across multiple tasks and models. Reviewers commend the novelty and importance of the work in addressing a pertinent challenge in LLM applications. However, concerns are raised about the method's performance with limited training prompts, lack of clarity in experimental details, and the potential limitations and computational costs in broader applications beyond arithmetic reasoning. Despite these concerns, the majority of reviewers acknowledge the paper's structured presentation, strong experimental validation, and the authors' comprehensive responses during the rebuttal phase. Suggestions for improvements include more explicit discussions on method limitations, additional comparisons, and better clarity on certain methodological aspects like the use of negative sampling and handling computational costs. Overall, the positive feedback on the relevance, novelty, and potential impact of the work supports a weak accept recommendation, pending revisions addressing the outlined concerns.

**Justification For Why Not Higher Score:**

The paper lacks clarity in certain methodological aspects and requires more comprehensive discussions on limitations and generalizability.

**Justification For Why Not Lower Score:**

The overall positive feedback from the reviewers regarding the relevance, novelty, and potential impact of the work, combined with the authors' comprehensive responses during the rebuttal phase, supports a weak accept recommendation.

---

### Decision · Program_Chairs · 2024-01-16

Accept (poster)